# Mapping the Land Use Changes in Cultivation Areas of Maize and Soybean from 2006 to 2017 in the North West and Free State Provinces, South Africa

Siphokazi Ngcinela [1,2,*], Abbyssinia Mushunje [1], Amon Taruvinga [1], Shelton Charles Mutengwa [3] and Samuel Tlou Masehela [4]

1   Department of Agricultural Economics and Extension, University of Fort Hare, Alice 5700, South Africa; amushunje@ufh.ac.za (A.M.); ataruvinga@ufh.ac.za (A.T.)
2   Directorate: Biodiversity Research Assessment and Monitoring, Kirstenbosch Research Centre, South African National Biodiversity Institute, 99 Rhodes Drive, Newlands, Cape Town 7735, South Africa
3   Department of Agronomy, University of Fort Hare, Alice 5700, South Africa; cmutengwa@ufh.ac.za
4   Biodiversity Risk Management, Biosafety and Alien Invasive Species, Department of Forestry, Fisheries and the Environment, Environment House, 473 Steve Biko Road, Pretoria 0083, South Africa; tmasehela@dffe.gov.za
*   Correspondence: ngcinelas@gmail.com or nqcinela@sanbi.org.za

**Abstract:** Land use practices face significant pressure due to increased demand and conflicting needs. Several factors contribute to this trend, such as the ever-increasing human population, the increased demand for food production, and the expansion of industrial and agricultural areas. This paper, focused on the cultivation patterns and investigating changes in land use of maize and soybean over time (i.e., both genetically modified and non-genetically modified) in two South African provinces. The objective was to determine whether there was a net increase or decrease in land cover age for these two crops between 2006 and 2017 in the selected study areas. Hence, the study utilized ArcGIS (10.8.1) software to quantify and map the land used for the cultivation of maize and soybean from 2006 to 2017 in Free State and North West provinces. The results show both provinces to have minimal expansion or change in cultivation areas for both maize and soybean between 2006 and 2017. We concluded that both maize and soybean cultivation areas in these provinces, did not expand beyond the current agricultural areas (space), and did not encroach onto new land areas. As a result, both maize and soybean do not currently pose a threat to the surrounding landscape (i.e., natural vegetation) and are not in direct competition with other neighboring land use practices. We recommend that data on the annual planting or cultivation area be consistently gathered, analyzed, and mapped to monitor any alterations that could influence the current findings. This will also assist with any land use planning and management practices.

**Keywords:** cultivation; genetically modified crops; land use; maize; soybean

## 1. Introduction

### 1.1. Global Overview of GM and Non-GM Crops Cultivation

One of the biggest challenge the world faces is how to feed an increasing population, which is expected to increase to 9.7 billion by 2050 [1,2]. It is expected that this will pose a challenge regarding how agricultural inputs match the outputs to keep up with demand. Agricultural food production will have to increase faster than the population growth [3,4]. This points to two things: (1) there will be an increased demand for arable land to expand the cultivation areas of various food crops [5], and (2) market and household crops that can generate more yield over the same or smaller cultivation areas will become a necessity [6]. All this will occur in the backdrop of other land use demands and competing needs, including conservation priorities for farmland biodiversity and ecosystem services/functions [7]. Thus far, it is well documented that agricultural practices such as crop production and

livestock rearing are some of the major contributors to land use change, often resulting in the deterioration of natural habitats [5]. Agriculture is said to be the single most important driver of biodiversity loss [5,6], alongside direct and indirect impacts of climate change, pests, and diseases [3,4].

It has been noted that the use of new technologies, including those that encompasses genetically modified (GM) or biotech crops, presents one potential avenue to accelerate the breeding of new crop varieties. This can assist farmers and agricultural systems in adapting to rapidly changing physical growing conditions and global markets [3,4,8]. Furthermore, it is emphasized that most of these crops are already widely used, over the last twenty years, in both industrialized and developing countries [9]. This is further supported by growing evidence indicating the continuous growth in GM crop production and the cultivation area [8–11]. For instance, in 2018, it was reported that the area planted globally reached about 191.7 million hectares [10,12]. In addition, about 87% and 95% of maize and soybean cultivated in South Africa (SA), respectively, is genetically modified.

Considering this, global trends indicate that there has been minimal expansion of agricultural land in the last few decades. However, upon the introduction of the technology, a slight decrease in agricultural land use area has been observed. This can be attributed to higher productivity in small areas of land [6]. Cropland is growing marginally in some parts of the world and pastureland is experiencing a slight decline [5,6]. However, both crops and livestock output are on the rise [12]. This suggests that in most regions around the world, agriculture has stayed within the space of agricultural land without encroaching on other areas. Furthermore, the literature also reveals a positive relationship between farmland and agricultural output [13]. While there is a consensus that the world human population is accelerating rapidly and is expected to increase to 9. 7 billion by 2050, there is no consensus on how this will influence other land uses and/or farm biodiversity. This includes uncertainties about whether there will be increased cultivation of land for sustainable food production and the sustainable utilization of other natural resources.

Simultaneously, significant gaps in data persist when it comes to accounting for various competing land use practices, particularly in the African continent. This continues to be a hindrance to tracking various impacts on biodiversity and the environment, as well as proper land use planning, management, and policy implementation.

In South Africa, there is little scientific evidence within the agricultural practices and management landscape to show interactions and/or tradeoffs between the different crops. The gains and losses in the cultivation area (land) in South Africa has not been clearly determined since the introduction of GM crops. It is therefore important to explore and understand how different crops are gaining or losing cultivation land over time, or how they interact with each other within cultivation areas. As a result, the main aim of this study was to map all cultivation areas (total land cover) for the three GM crops (maize, soybean and cotton) within the proximity of non-GM crops/natural vegetation and assess any trade-offs in terms of the land use ratio from 1998 to 2019 in North West, Mpumalanga, Limpopo, Gauteng, Free State, KwaZulu-Natal and Western Cape provinces. We wanted to establish land use trends with respect to both GM and non-GM cultivation over a period of time. However, due to insufficient spatial data for both maize and soybean across all provinces, we only quantified and mapped land use trends in the cultivation areas for maize and soybean in the Free State and North West provinces, for the years 2006 to 2017. These are well-known and documented areas where both crops are mainly cultivated [14]. We explored how each crop gained or lost its area of cultivation over time. Here, it was important to address the data and knowledge gap on land gains and/or losses associated with the cultivation of both crops, and to establish any implications for other crops and the natural vegetation. It is important to mention that the data used in this study (and analysis) did not make a distinction between the total area cultivated for GM (soybean and maize) versus that of non-GM crops (conventional) for the respective provinces.

### 1.2. Overview of GM Crops Cultivation in South Africa

#### 1.2.1. Status of GM Crops in SA

South Africa has been ranked as one of the top ten developing countries (position nine) for adopting and cultivating GM crops [11,14]. To date, the country grows three major GM crops: maize, cotton and soybean on a combined 2.7 million hectares [15]. The three crops, maize, cotton and soybean, were first planted in SA in 1997, 1998, and 2001, respectively [11,14]. The two sections below only cover the literature on maize and soybean since cotton has not been covered in this study for the investigation of land use change or patterns.

#### 1.2.2. GM Soybean

Genetically modified soybean has been planted in SA since 2001 [11,16]. Over twenty-one years of GM crop commercialization in SA, GM soybean has been noted as the fastest growing field crop compared to maize and cotton [11]. However, there was a slight decline in its production in the 2015/2016 season due to drought [11]. In 2017, about 787,200 hectares of this crop were cultivated in SA [11]. In addition, farmers in South Africa are currently producing the two genetically modified traits, namely the Roundup Ready (GTS 40-3-2) trait, which was approved in August 2001, and the Intacta® Roundup®Ready2 Pro (Intacta® RR2 Pro), which was approved July 2021 [17]. Moreover, since the introduction of GM soybean in SA, high production levels have been achieved [18]. The rise in production yields of soybean is driven by the growth of the livestock and poultry sectors, driven by the rise in disposable income among the population, which in turn increases the demand for animal protein [16].

Additionally, the rising yields are supported by the favorable agricultural environment policy backing the commercialization and use of agricultural biotechnologies [16]. Soybean is produced throughout the country, but significant production takes place in Free State and Mpumalanga provinces due to their favorable soils and climatic conditions [18]. This significant increase is further driven by farmers in South Africa being able to rotate soybean with other grains, in particular maize, during the planting period, as this also serves as a means of profit maximization [16]. It is worth noting that the country is the largest importer of soybean oilcake in Sub-Saharan Africa, accounting for an average of 72% of import demand [18]. This is due to the increased crushing capacity and crop rotation advantages of soybeans [19].

#### 1.2.3. GM Maize

Since the first introduction of GM maize in South Africa in 1998, the national GM maize area has gradually increased, reaching a level of around 90% of the total maize area in 2016/17 [20]. Majority of the maize produced in South Africa is GM, therefore, most of the surplus maize available for exports is also predominantly GM [21]. In 2017, it was estimated that the area under GM maize was 85% commercial, against the 90% recorded in 2016 [11,22]. As a result, the total area cultivated under maize was 1.96 million hectares in 2017, compared to the 2.3 million hectares in 2016.

The total commercial GM maize area in SA is estimated at 93.8%. With the inclusion of smallholder farmers, the national GM percentage dropped to 83.6% for the 2016/2017 production season [20,21]. Moreover, the general maize production trend is skewed towards GM maize, and this is mainly explained by the economic benefits linked to GM maize [20,21]. In addition, GM maize production has been associated with improved enterprise competitiveness through the impacts it has on yields, the cost of production and product prices [21,23]. The stacked GM trait (Br) for both yellow and white maize constitutes the highest share of the total commercial maize area at 76% for white maize and 66% for yellow maize [21]. Moreover, the current GM maize traits produced by farmers in South Africa are herbicide tolerant (Ht), *Bacillus thuringiensis* (Bt) and a combination of the two stacked genes (Br) [17]. Furthermore, it has been projected that the total supply of white and yellow maize for the 2018/19 marketing season was 8,993,375 tons and 6,970,159 tons,

respectively [17]. As a result, the total demand for white and yellow maize, domestic and for export, was estimated at 7,300,000 tons and 5,658,000 tons, respectively, for the 2018/19 marketing season [17].

## 2. Study Methods

### 2.1. Description of Study Sites

This study was carried out for the Free State and North West provinces only (Figure 1), due to the data that were available for these provinces. The Free State province is predominantly rural farmlands, mountains, goldfields, and dispersed towns with a population of 2,834,714. About 90% of the province is under crop cultivation. The province produces approximately 34% of the total maize production of South Africa, 37% of wheat, 53% of sorghum, 33% of potatoes, 18% meat, 30% of groundnuts, and 15% of wool [24]. The total land size cultivated in the Free State in 2021 was 1,325,000 and 365,000 hectares, for maize and soybean, respectively [25].

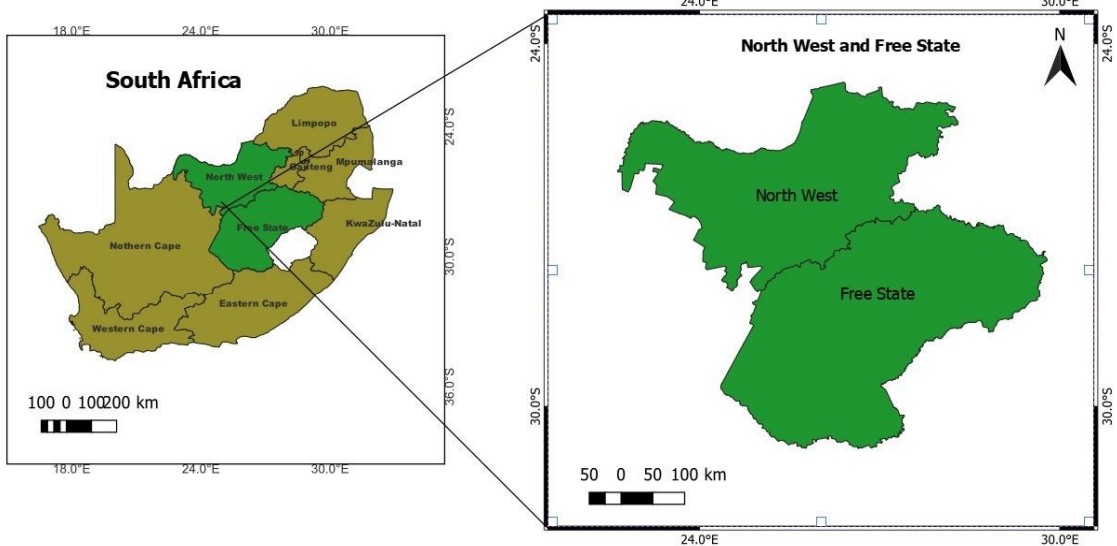

**Figure 1.** Illustrates the two study areas, the North West and Free State provinces of South Africa.

The North West province has a population of 3,748,436 and is well known for cattle farming, with mixed farming land. Maize and soybean are the most important crops, and the province is a major producer of white maize in the country [24]. Also, in the province, there is a mining sector which is a major contributor to the economy. The province produces a quarter of South Africa's gold, as well as granite, marble, fluorspar, and diamonds. The total land size cultivated in the North West in 2021 was 573,000 and 69,500 hectares for maize and soybean, respectively [25].

### 2.2. Data Source

A longitudinal research design was used to assess the trend analysis of maize and soybean cultivation in the Free State and North West provinces over a period of 12 years. Secondary data were used for the total area cultivated under maize and soybean from 2006 to 2017, for the Free State and North West provinces. Both quantitative and qualitative data were obtained in the form of GIS shape files. This allowed us to investigate and establish which crop (maize and soybean) gained or lost cultivation area/land cover over a period of 12 years.

Total cultivated area data in shapefiles for maize and soybean were obtained from the Crop Estimates Committee (CEC), a Branch of the Department of Agriculture, Land Reform and Rural Development (DALRRD).

### 2.3. Analytical/Mapping Method and Limitations

Arc GIS software (10.8.1) was used to generate maps and to calculate the total area cultivated under maize and soybean between 2006 and 2017. The aim was to establish which crop gained or lost land cover over time and to ascertain whether these crops are replacing each other, other agricultural crops, or natural vegetation in terms of cultivated area over a specific period.

**Note:** Software limitations: out of the total area that was calculated or shown by the maps, the software was not able to tell or distinguish how much area was under GM (soybean and maize) or non-GM crops for each respective province. Due to having a high proportion of GM maize and soybean, 85% and 95%, respectively, we assumed that the majority of cultivated or land cover for these crops is GM in SA.

### 3. Results

The results in Figure 2 show the land use trends (cultivation area patterns) for maize and soybean from 2006 to 2017 for the Free State province. In 2006, maize cultivation was not as high as in subsequent years. The generated maps show that there was a lower concentration of maize in 2006 as compared to previous years. This implies that in 2006, maize was not encroaching on areas that were not previously used for maize production. Therefore, maize stayed within the space allocated for maize production. At the same time, the cultivation area for maize was mostly in the Northern part of the province from 2006 to 2017. Soybean cultivation was concentrated on the Eastern side of the province (2006 to 2017). This means that maize and soybean were grown in and restricted to different regions of the province. As a result, these two crops were not primarily cultivated together as anticipated, despite their well-known tendency to be planted together [26]. When looking at both crops, the 2006 to 2009 data (maps) indicate a growth in the area under soybean. Subsequently, the area under soybean has remained constant or decreased. There was a lack of clarity in the maps from 2010 to 2017 regarding the shift in soybean cultivation as a result of maize over-shadowing it. From 2010, there was a shift in the production of maize towards the western part of the province, reflected by the darkening of areas in the 2010 maps onwards, indicating that there was a higher concentration of maize in this area. This suggests that maize encroached on areas that were not previously used for maize cultivation. Moreover, it appears that 2016 was an outlier year, as maize production was not as concentrated compared to other years. This can be attributed to the drought years that might have resulted in a decrease in the production area for maize in the province [27,28].

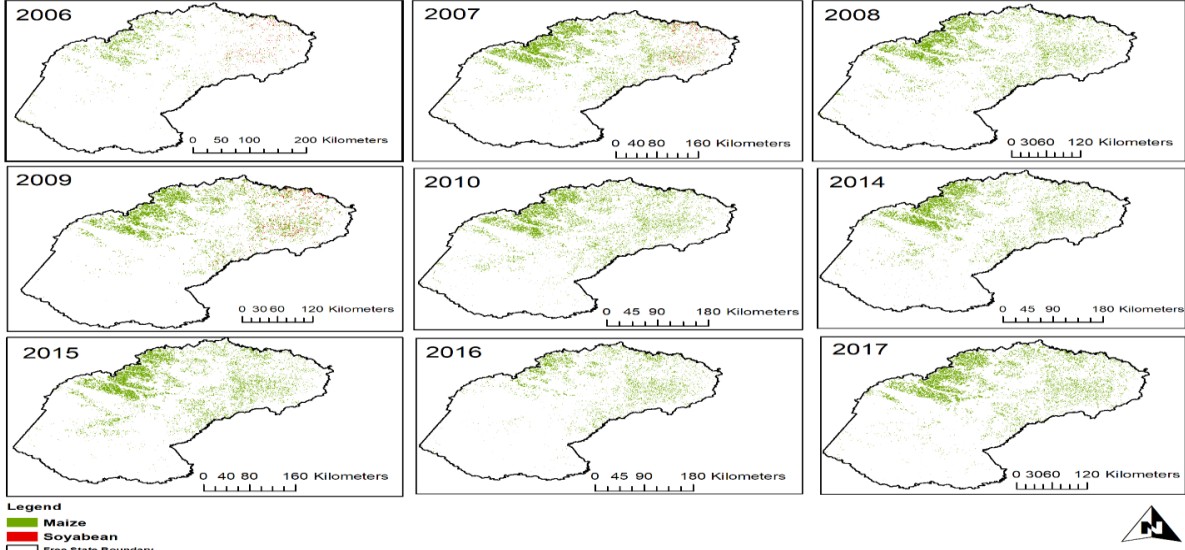

**Figure 2.** Cultivation patterns for maize and soybean in the Free State province, South Africa, from 2006 to 2017.

Table 1 illustrates the total hectares planted on in the Free State province, as well as the percentage of change in the area (gained or lost) for both soybean and maize between 2006 and 2017. The results show that there were fluctuations in the production of both crops from 2006 to 2017. Nevertheless, the maps (Figure 2) did not clearly illustrate significant distinctions in the cultivated area for these crops. However, when considering the percentage change for both maize and soybean, the difference was minimal. This implies that there was a marginal increase in the cultivation area for both maize and soybean from 2006 to 2017. Apart from 2016 for maize, which was a drought year [27], there was a marginal increase in the area cultivated for both crops (Table 1). Although there was minimal expansion in the cultivation area of maize and soybean crops, it is not clear or known if these two crops are replacing each other in terms of the area cultivated. Similar findings of a slight increase in the area cultivated under GM crops were reported in other studies [6,12,13]. Also, there was no clear indication or evidence of vegetation clearing from natural areas to plant "new" GM crops. At present, the findings suggest that both maize and soybean do not pose a threat to biodiversity and the adjacent natural habitats (vegetation).

**Table 1.** Total area and percentage of change in area of maize and soybean from 2006 to 2017 in the Free State Province, South Africa.

| Class | 2006 Area (Ha) | 2007 Area (Ha) | 2008 Area (Ha) | 2009 Area (Ha) | 2010 Area (Ha) | 2014 Area (Ha) | 2015 Area (Ha) | 2016 Area (Ha) | 2017 Area (Ha) |
|---|---|---|---|---|---|---|---|---|---|
| Maize | 692,197.23 | 1,121,333.28 | 1,136,310.69 | 980,871.08 | 1,201,651.83 | 1,144,689.91 | 1,259,515.36 | 691,437.84 | 1,173,676.42 |
| Soybean | 49,165.15 | 44,973.48 | 40,375.24 | 67,748.82 | 51,919.29 | 224,395.15 | 301,315.72 | 189,547.22 | 308,586.50 |
| Class | | %Change 2006–07 | %Change 2007–08 | %Change 2008–09 | %Change 2009–10 | %Change 2010–14 | %Change 2014–15 | %Change 2015–16 | %Change 2016–17 |
| Maize | | 61.99 | 64.16 | 41.70 | 73.59 | 66.57 | 81.95 | −0.11 | 69.55 |
| Soybean | | −8.53 | 17.88 | 37.79 | 5.60 | 356.41 | 512.86 | 3755.27 | 527.65 |

Figure 3 shows the mapped areas for land use trends (cultivation area patterns) of maize and soybean from 2006 to 2017 in the North West province. The results indicate that soybean was mostly concentrated in the eastern side of the province, while maize was mostly concentrated on the southern side of the province. This means that maize and soybean were largely grown in and restricted to different regions of the province. The results further showed that soybean cultivation was minimal relative to the maize cultivation area from 2006 to 2017. From 2011 to 2017, the color of maize areas in the respective maps became darker, indicating that there was a higher concentration of maize moving to the eastern side of the province. Suggesting that maize cultivation was increasing during these years (2011 to 2017). This indicates that the province yielded a higher maize output compared to soybeans, possibly due to high demand and/or favorable weather conditions. Additionally, starting from 2012, maize cultivation began expanding towards the northern region of the province, implying a growing trend of maize cultivation encroaching into areas not previously utilized for this purpose. What is not clear from the analysis and results is whether those areas were under natural vegetation or other agricultural practices. From these maps, the study only detected and observed an increase in the concentration of maize in those areas. Also, the maps do not show the concentration of maize in those areas and whether it was GM or non-GM. The area cultivated under these crops was calculated to show how much change has occurred over time.

After mapping the land use trends between the cultivation of maize and soybean (Figure 3), we calculated the area cultivated under these crops from 2006 to 2017. Looking at the percentage of change for both maize and soybean, the difference was minimal. This indicates that both crops in the province remained restricted to the agricultural land, without encroaching much on other new areas. This observation is similar to that made for the Free State province for both maize and soybean. These findings are also in line with those by [13], who reported that the global cropland grew only slightly or slowly from

1961 to the early 1990s, and did not increase thereafter, in spite of the rapid increase in the agricultural output. [13] further projected that the expansion in cropland in the future would be limited. These findings concur with [5], who projected that the world food demand would be met by the supply of agricultural products (i.e., the productivity) with just a marginal expansion of agricultural land. These studies highlighted that agricultural growth would come from the actual productivity rates (i.e., increased yields), and not production generated through land expansion for more crop cultivation [5,13]. This study, like the findings in the Free State province, could not determine if these crops are substituting other agricultural crops in the North West province. The data presented in Table 2 indicate a slight uptick in maize cultivation area from 2006 to 2012 in the province, followed by a modest decrease after 2012. Conversely, from 2014 to 2017, there was a gradual expansion in the area allocated for soybean cultivation. This trend could be attributed to the growing adoption of soybean driven by increased market demand [25,29].

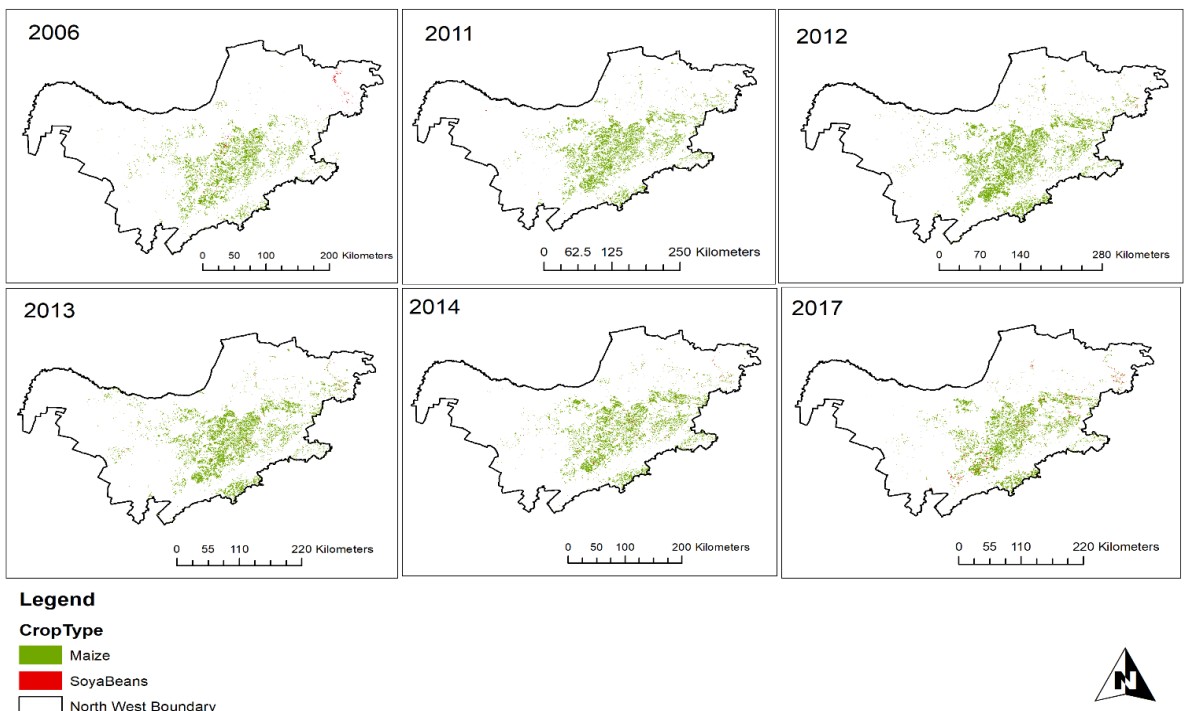

**Figure 3.** Cultivation patterns for maize and soybean in the North West province, South Africa, from 2006 to 2017.

**Table 2.** Total area and percentage (%) change in area for both maize and soybean from 2006 to 2017 in the North West province, South Africa.

| Class | 2006_Area (Ha) | 2011_Area (Ha) | 2012_Area (Ha) | 2013_Area (Ha) | 2014_Area (Ha) | 2017_Area (Ha) |
|---|---|---|---|---|---|---|
| Maize | 494,891.44 | 727,613.54 | 823,167.56 | 763,108.06 | 669,041.77 | 612,798.63 |
| Soybean | 13,648.13 | 2660.02 | 3736.84 | 9230.22 | 15,739.41 | 31,321.46 |
| Class | | %Change 2006–11 | %Change 2011–12 | %Change 2012–13 | %Change 2013–14 | %Change 2014–17 |
| Maize | | 47.02 | 66.33 | 54.19 | 35.19 | 23.82 |
| Soybean | | −80.5 | −72.60 | −32.40 | 15.32 | 129.49 |

Table 2 is used to present and illustrate the changes from 2006 to 2017 for the provinces that are not easily detectable using Figure 3.

## 4. Discussion

Findings drawn from both provinces mainly indicate that the cultivated area for the two crops (maize and soybean) had marginal fluctuations or shifts. This accounts for the minimal increase between 2006 and 2017. Although there was a minimal expansion in the cultivation area of maize and soybean in both provinces, it remained unclear whether these two crops were substituting each other in terms of cultivated area. Furthermore, there was a no clear indication of either crop encroaching on natural areas. Here, it is important to note that the results indicate maize and soybean were grown in and restricted to certain or different regions of the Free State province, that is, the north and west parts of the province for maize and the east part for soybean.

This was primarily driven by several factors, including the availability of agricultural land suitable for each crop. These crops were planted in different regions to mitigate yield reduction, given their differing competitive abilities. Additionally, separate management practices were implemented for each crop due to their distinct agricultural requirements, despite the occasional practice of planting them together or adjacent to one another [11,15,26]. Furthermore, soybean is predominantly utilized as a rotational crop with maize [5,10,11,15,26,27]. Further contributors include farmers' openness to experimenting with new varieties, adopting improved farm management techniques, taking advantage of the expansions in the markets, and striving to maximize profits [13,19,26]. Additionally, in the case of the Free State province (refer to Figure 2), it seems that the year 2016 stood out as an anomalous year, with maize cultivation appearing to be less concentrated compared to other years. This anomaly could be attributed to the drought experienced during that period, as 2016 was characterized by drought conditions, likely leading to a reduction in maize production area in the province—[5,10,27].

Similar results to those of the Free State provinces were noted for the North West, whereby maize and soybean were grown in and restricted to different regions of the province, that is, the south and east sections of the province, respectively.

The rationale behind planting these crops in separate regions may stem from the availability of agricultural land suited for each crop, as well as the convenience of managing them individually due to their distinct agricultural practices. [5,9,11,26]. Also, farmers often have their own plans and targets, so this dictates their priorities in their crop planting preferences. The findings further suggest that for the period of 2006 to 2017, both crops in the two provinces remained restricted to the agricultural land, without encroaching much on other new areas. However, the results do not definitively indicate whether these crops are displacing other agricultural crops. Moreover, there was no indication of clearing natural areas to plant new crops (GM or non-GM). This suggests that both maize and soybean do not currently pose a threat to the surrounding landscape, including natural vegetation, and are not in direct competition with other neighboring or competing land use practices. From 2011 to 2017 (see Figure 3) the area cultivated with maize increased, and we attribute this change or shift to high demand and/or favorable weather conditions for maize compared to soybean [20,21,26,30].

## 5. Conclusions

The findings show that both maize and soybean in the two provinces have thus far remained within their agricultural areas, with no substantial evidence to indicate that they are either replacing other crops in these areas or encroaching on neighboring areas (e.g., natural vegetation). This holds true even within the context of introducing GM crops. These results are consistent with findings from other areas or regions, indicating minimal growth or shifts in the cultivation of such crops, albeit with increased productivity such as crop yield. Given these findings, we concluded that both maize and soybean, do not currently pose a threat to the surrounding landscape and are not in direct competition with other neighboring land use. Therefore, their cultivation and management (currently) remain within the desired agricultural management practices without threatening the valuable natural areas of South Africa.

## 6. Recommendations and Policy Insights

- There is an urgent need for more robust real-time spatial data for these crops (and others) to allow for more extensive analysis and monitoring the trends of these crops regularly in terms of their interactions at a landscape level with their counterparts and the surrounding landscapes. Hence, having access to reliable data would be crucial for conducting a comprehensive trend analysis of maize and soybean cultivation areas, potentially on a national scale, both before and after the introduction of GM events/traits. This will be essential for future agricultural land planning, prioritizing conservation efforts, and addressing other land use and development requirements. Hence, it is crucial for value chain stakeholders to ensure the availability of adequate data.

- There is a strong possibility that these crops may continue to be confined to their current agricultural land, implying that land currently unused by these crops would not face threats from their production. However, it is uncertain whether the same scenario would apply to other provinces. Therefore, it is crucial to prioritize similar studies in other provinces, considering the necessity for comprehensive real-time spatial data for various crops.

- Regularly assessing changes in the cultivation area, such as every five years, is crucial to determine whether any increases in the cultivation of these crops would continue to not pose a threat to the surrounding landscape. This includes vital biodiversity areas or neighboring farmlands with significant ecosystem functions or services. Such assessments are necessary for informing future agricultural land planning and environmental management decisions by policy makers.

**Author Contributions:** Conceptualization: The following authors conceptualized the study: S.N., S.T.M., A.M., A.T. and S.C.M. Methodology: S.N., S.T.M., A.M., A.T. and S.C.M. Software: S.N., S.T.M. Validation: S.N., S.T.M., A.M., A.T. and S.C.M. Formal analysis: S.N. and S.T.M. Investigation: S.N. and S.T.M. Data curation: S.N. and S.T.M. Original draft preparation: S.N. Writing: S.N. Review and editing: S.T.M. Visualization: S.T.M. Supervision: S.T.M., A.M., A.T. and S.C.M. Project administration: S.N., A.M. and S.T.M. Funding acquisition: S.T.M. All authors have read and agreed to the published version of the manuscript.

**Funding:** This research was funded through South African National Biodiversity Institution (SANBI).

**Data Availability Statement:** The data that support the findings of this study is available from the corresponding author.

**Acknowledgments:** The paper was developed from a doctoral study conducted at the University of Fort Hare, South Africa. Data sourcing was carried out through the respective government departments, including: (1) the Crop Estimates Committee (CEC), a Branch of the Department of Agriculture, Land Reform and Rural Development (DALRRD); and (2) Geoterra Image (Pty) Ltd. for their assistance in providing the respective data sets. Gratitude to Fhatani Ranwashe from SANBI who assisted with the ArcGIS analyses. Appreciation to Sinikiwe Simakani for independent review and editing of the paper.

**Conflicts of Interest:** No potential conflicts of interest to be reported/disclosed.

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
