# Peer review of "Mapping the Land Use Changes in Cultivation Areas of Maize and Soybean from 2006 to 2017 in the North West and Free State Provinces, South Africa"

_agronomy, doi:10.3390/agronomy14051002_

Round 1

Reviewer 1 Report

Comments and Suggestions for Authors

Abstract: Why did you choose only maize and soybeans in your study? Does the land cultivate those crops only for the whole year? Please show your importance before starting your objectives, methods, and results in the abstract section. What are your study recommendations for future study or policy maker or decision maker? Why did your study cover 2006 to 2017? Why not analyze the land cover change in recent years, such as 2023, 2022, or 2021? It's very old data; perhaps your study results do not address the recent patterns. The methodology is not clear and is very short in the abstract; please write clearly what methods and modeling were followed in GIS.

Keywords: suggest only five keywords, keep only the important 5, and remove others

Introduction

Line 39 to 41: I am not clear and convinced; please paraphrase and revise the sentence.

You do not need to analyze a separate literature review in the introduction part. I suggest modifying the heading (remove heading 2) and revising the writing.

What is the importance of Figure 1 in your study? Where did you analyze in the introduction part? This is not an updated figure; I suggest keeping updated data.

Methods

Remove line 191-192.

Section 3.1 Insert study area figure showing location, major city.

What kind of both quantitative and qualitative data was obtained in GIS shapefiles? Describe it.

What kind of data for maize and soybean was obtained from the Crop Estimates Committee? Is it annual crop production data or a cultivated GIS file?

Enlarge Figure 3 and 4. It is difficult to notice the soybean cropping pattern.

How did you map Figures 3 and 4? What is the process? Please describe in the method sections by showing the steps (diagram/ flow chart).

Why the mapping of the time interval is not the same in Figures 3 and 4? Suggest to make same time interval. You can follow a two-year interval. The north arrow should always be inside the map.

Suggest making a figure showing a loss and gain area diagram.

Discussion and conclusion

There is not enough discussion to prove your study is good and noble. You should review and cite several similar studies to support your results. For example, the cropland decreased significantly in 2015 and 2016 in Free State, but you did not analyze it in the discussion part. I suggest reviewing and citing at least ten pieces of literature in this section.

I suggest keeping a separate section for the conclusion.

Recommendation

I did not get the clear meaning of your recommendation and suggested policy. Please revise it.

Author Response

Greetings,

Reviewer 2 Report

Comments and Suggestions for Authors

The introduction must be more consistent, more citations from the international bibliography.

Chapter 2, the one with the literature review, should be moved to the first part.

The methodology is not at all clear.

What is actually the source of the data? Did you collect them from the field or did you take them from an institution?

It is only understood that you made some maps with those data.

Made a graph with the work flow so that we can more easily follow the steps you followed.

Results: Move Figure 2 after the Description of Study Sites.

Make some graphs after Table 1; they are easier to follow.

Enlarge the maps in Figure 3. You can't understand anything.

If you want to make a comparison between the same maps, I recommend using the same scale.

What I wrote above is also valid for figure 4.

In case of discussion, I recommend you compare your results with other results obtained in similar studies. Try to highlight whether the patterns in your case are similar to patterns from other scenarios.

Introduce more international bibliography; in this way, you increase the visibility of your article.

Treat the conclusions chapter separately, taking information from the Recommendations & Policy Insights subchapter.

I do not think the paper can be published in this form, which is why I will give it a major review.

Author Response

Greetings,

Reviewer 3 Report

Comments and Suggestions for Authors

1. What knowledge does this article offer readers? It is advisable to clarify this aspect explicitly.

2. The examination results reflect the author's diligent and rigorous research, deserving commendation. However, the relevant discussion preceding the conclusion requires a focused review rather than a direct presentation of the conclusion. Otherwise, this paper may seem more like an informal academic work than a meticulous scholarly article.

3. The provided reference materials are inadequate, and their representativeness is concerning. Some references are not cited in the main text, while others mentioned in the main text are omitted from the reference list. This aspect requires improvement.

4. The writing style of the article appears somewhat casual, and there is a need for a more rigorous examination and confirmation of the relationship between the proposed indicators and research hypotheses.

Author Response

Greetings,

Round 2

Reviewer 1 Report

Comments and Suggestions for Authors

Dear authors, 

I found most of the comments and suggestions are addressed. However, all the figures are not readable. I strongly recommend to revise it with readable text, scale (follow uniformity scale and readable), and legend with an enlarged figure size. Can you read out Figure 1? Still suggest to review and analyze more papers, your citations are not enough. Mostly cite at least 10-15 papers in the discussion part. 

Reviewer 2 Report

Comments and Suggestions for Authors

Explain This image. You have the same region, but different years.  If the spatial element is the same, why do you have different scales? Keep the same scale.

When I said enlarge the images, I meant enlarge them and improve the quality.

Add a workflow diagram for your Methodology.
